# Wastewater Metavirome Diversity: Exploring Replicate Inconsistencies and Bioinformatic Tool Disparities

**DOI:** 10.3390/ijerph22050707

**Published:** 2025-04-30

**Authors:** André F. B. Santos, Mónica Nunes, Andreia Filipa-Silva, Victor Pimentel, Marta Pingarilho, Patrícia Abrantes, Mafalda N. S. Miranda, Maria Teresa Barreto Crespo, Ana B. Abecasis, Ricardo Parreira, Sofia G. Seabra

**Affiliations:** 1Global Health and Tropical Medicine (GHTM), Associate Laboratory in Translation and Innovation Towards Global Health (LA-REAL), Instituto de Higiene e Medicina Tropical (IHMT), Universidade NOVA de Lisboa, 1349-008 Lisboa, Portugal; andretomasdossantos@gmail.com (A.F.B.S.); victor.pimentel@ihmt.unl.pt (V.P.); martapingarilho@ihmt.unl.pt (M.P.); patriciaabrantes@ihmt.unl.pt (P.A.); mafaldansmiranda@ihmt.unl.pt (M.N.S.M.); ana.abecasis@ihmt.unl.pt (A.B.A.); ricardo@ihmt.unl.pt (R.P.); 2cE3c—Centre for Ecology, Evolution and Environmental Changes & CHANGE—Global Change and Sustainability Institute, Faculdade de Ciências, Universidade de Lisboa, Campo Grande, 1749-016 Lisbon, Portugal; msnunes@ciencias.ulisboa.pt; 3CIIMAR/CIMAR-LA, Centro Interdisciplinar de Investigação Marinha e Ambiental, Universidade do Porto, Terminal de Cruzeiros do Porto de Leixões, Av. General Norton de Matos, s/n, 4450-208 Matosinhos, Portugal; andreia.silva@ciimar.up.pt; 4iBET, Instituto de Biologia Experimental e Tecnológica, Apartado 12, 2781-901 Oeiras, Portugal; tcrespo@ibet.pt; 5ITQB, Instituto de Tecnologia Química e Biológica António Xavier, Universidade Nova de Lisboa, Av. da República, 2780-157 Oeiras, Portugal

**Keywords:** wastewater, metagenomic analysis, environmental surveillance, next generation sequencing

## Abstract

This study investigates viral composition in wastewater through metagenomic analysis, evaluating the performance of four bioinformatic tools—Genome Detective, CZ.ID, INSaFLU-TELEVIR and Trimmomatic + Kraken2—on samples collected from four sites in each of two wastewater treatment plants (WWTPs) in Lisbon, Portugal in April 2019. From each site, we collected and processed separately three replicates and one pool of nucleic acids extracted from the replicates. A total of 32 samples were processed using sequence-independent single-primer amplification (SISPA) and sequenced on an Illumina MiSeq platform. Across the 128 sample–tool combinations, viral read counts varied widely, from 3 to 288,464. There was a lack of consistency between replicates and their pools in terms of viral abundance and diversity, revealing the heterogeneity of the wastewater matrix and the variability in sequencing effort. There was also a difference between software tools highlighting the impact of tool selection on community profiling. A positive correlation between crAssphage and human pathogens was found, supporting crAssphage as a proxy for public health surveillance. A custom Python pipeline automated viral identification report processing, taxonomic assignments and diversity calculations, streamlining analysis and ensuring reproducibility. These findings emphasize the importance of sequencing depth, software tool selection and standardized pipelines in advancing wastewater-based epidemiology.

## 1. Introduction

Wastewater viral monitoring provides a tool for the early detection and tracking of pathogenic viruses and their genetic diversity circulating in human and animal communities excreting to that wastewater [1,2,3,4]. Several studies in poliovirus [5,6], and SARS-CoV-2 [7,8], highlight the value of this approach in epidemiological surveillance [8,9,10]. Viral detection can also be applied to assess the efficacy of wastewater treatment in removing viral pathogens [11,12].

Targeted methods, such as amplification using specific PCR primers, are typically used to detect specific viruses in wastewaters (e.g., SARS-CoV-2 [8], polio virus [6], Influenza virus [13], Mpox virus [14], mastadenoviruses [15], dengue virus [16,17] and Chikungunya virus [16,18]). Metagenomics offers a broader view, uncovering viral diversity in wastewater in an untargeted way [9,16,19,20,21]. It enables characterizing the diversity of known and emerging viruses [9,16,19,20,21,22,23,24,25,26,27], as well as discovering new uncharacterized ones [24,28]. A metagenomic approach poses significant challenges when sequencing very complex matrices like wastewater. The vast proportion of genetic material belongs to bacteria, and, within the small proportion of viruses, the vast majority comprises bacteriophages (phages) [2,8,29,30,31,32] and plant viruses present in human/animal feces [2,32].

Metagenomic studies have demonstrated that viral reads in wastewater samples often constitute a very small proportion compared to the total reads, highlighting the challenges of detecting viruses amidst a microbial community dominated by bacteria [33,34,35].

Total RNA or DNA sequencing has been used to characterize the virome of wastewaters [2,32]. Metabarcoding, typically based on the analysis of 16S rDNA (for bacteria community characterization) or using the COI sequence (for eukaryotes), is not suitable for viruses due to the lack of universal genetic markers, while, and as an alternative, one can target the amplification of specific viral families [19,20,25]. For a more comprehensive analysis of the virome, an alternative is to use enrichment methods, for example with viral probes, and random primer amplification [4,24,36,37]. However, enrichment methods can distort viral abundance profiles [32,38,39,40], calling for standard protocols and benchmarking. Biases may also be introduced during sample collection and processing [38,41,42]. One primary bias arises from sampling methods, such as grab samples, which capture only a snapshot of the wastewater at a single time point and may not reflect variations in the viral or microbial load throughout the day. Composite sampling over longer periods can mitigate this but introduces its own challenges, such as the degradation of sensitive bio-molecules like RNA during collection or storage [43]. Another source of bias is heterogeneity in wastewater composition, influenced by population density, industrial discharge and environmental factors like rainfall, which can dilute samples or introduce contaminants [44]. Sample handling and processing factors, for instance, inefficient concentration methods, may preferentially recover some microorganisms over others [34]. Lastly, biases can stem from analytical techniques, such as PCR amplification bias, or reagent contamination, which can introduce false positives or obscure low-abundance taxa [45]. Recognizing and mitigating these biases through careful sampling design, robust protocols and appropriate controls is critical for reliable wastewater-based surveillance.

In metavirome analysis, where the focus is on viruses—which often have high mutation rates and genetic diversity—starting with high-quality data is essential. Raw sequencing data often contains errors, low-quality reads and sequencing artifacts. The first key step is to filter them out and trim sequencing adapters. The following step, metagenomic assembly, the process of the reconstruction of genomes from the smaller DNA segments (sequencing reads), presents challenges given the high diversity and often low representation of viral genomes in environmental samples. The assembly of genomes directly from sequencing reads without relying on reference genomes [46,47] is a particularly useful approach in metagenomics, where many organisms lack well-characterized reference sequences [48,49]. Taxonomic classification in viral metagenomics is carried out using two primary approaches—k-mer-based and alignment-based methods—to find matches with sequence databases.

K-mer-based tools [50,51,52] perform rapid matching of short sequence fragments to indexed reference databases, though they may lack sensitivity for novel or highly diverse sequences [53,54]. Alignment-based methods [55,56,57] are computationally demanding but offer greater precision by aligning sequences to known databases [58]. Comparative analyses often indicate that alignment-based tools yield higher accuracy for classifying diverse viral sequences, whereas k-mer-based methods remain advantageous for high-throughput datasets due to their speed [54,59].

In this study, we used metavirome sequences from wastewater samples, processed by site in triplicate and in pools, to analyze the consistency of several bioinformatic analysis pipelines/workflows available online in characterizing their viral taxonomic composition. We specifically aimed (1) to compare the viral occurrence and abundance obtained for each replicate and pool by each software pipeline; (2) to identify factors that may impact the variability among replicates and pipelines.

## 2. Materials and Methods

### 2.1. Wastewater Collection, Processing and Nucleic Acid Extraction

Samples were collected in April 2019 from two wastewater treatment plants (WWTPs) in Lisbon, Portugal. At each WWTP, three replicate samples were taken from four different sites (treatment steps), with a total of 32 samples (Figure 1). “Treatment steps” refers to distinct stages within each wastewater treatment plant (WWTP), namely, (1) influent (raw wastewater entry), (2) post-primary treatment (solids and sediments removal), (3) secondary treatment (biological treatment with microbial processes) and (4) effluent (final treated wastewater released into the environment). The collected water volumes were 1 L for the influent samples and 10 L for the others (Appendix A). The samples were immediately transported to the lab, stored at 4 °C and processed within 24 h for viral particle concentration using a series of methods, including organic flocculation, sedimentation, centrifugation and re-suspension, adapting a previously published protocol [9].

RNA extraction was carried out from 140 mL of the concentrated sample with QIAamp® Viral RNA Mini Kit (QIAGEN®, Hilden, Germany) following the manufacturer’s protocol. This extraction method allows recovering RNA but also DNA from samples. RNA was quantified and quality ratios were obtained for each extraction using NanodropOne. A subsample of the nucleic acid extractions of each of the three replicates from each site (treatment step) was taken, normalized into an RNA concentration of 50 ng/µL and then 1/3 of the volume taken from each of the replicates was used to create a pool for that site (Figure 1).

The pooled samples were all standardized to an RNA concentration of 50 ng/µL. Both replicates and pool from each site in a total of 32 samples (2 WWTP × 4 sites × 3 replicates + 1 pool), were further processed as detailed in the next section.

### 2.2. Sequence-Independent Single-Primer Amplification (SISPA), Viral Enrichment and Sequencing

The nucleic acid extracts were randomly amplified using the sequence-independent single-primer amplification (SISPA) protocol [25], which enables the analysis of diverse or unknown genomes. It is particularly useful in metagenomics and pathogen discovery, where prior knowledge of the genome sequence is unavailable. SISPA can amplify both RNA and DNA. RNA is typically reverse transcribed into complementary DNA (cDNA) before the amplification process, while DNA can be directly used. The workflow included the detection of both RNA and DNA viruses. To enrich for viral sequences, libraries were processed using the VirCapSeq-VERT capture panel (Roche, Switzerland), which includes approximately two million probes targeting 207 viral taxa that infect vertebrates, thereby facilitating the detection of viral sequences in complex samples [20]. Library preparation utilized the HyperCap Target Enrichment Kit and the HyperCap Bead Kit (Roche, Basel, Switzerland). Hybridization with VirCapSeq-VERT probes was performed at 47 °C for 20 h. Captured DNA was retrieved using magnetic beads (HyperCap Bead Kit, Roche) and subsequently cleaned. The bead-bound DNA was subjected to ligation-mediated PCR (LM-PCR) for further purification. The enriched libraries were sequenced on an Illumina MiSeq platform, generating paired-end reads using a 2 × 300 bp paired-end protocol. Negative controls without template were included in all reactions to exclude the possibility of contamination.

### 2.3. Bioinformatic Analysis

The study applied three bioinformatic pipelines available online—Genome Detective (GD) [61], CZ.ID [62] and INSaFLU-TELEVIR [63], and Trimmomatic + Kraken2 run on Galaxy [64]—to analyze 32 sequenced samples (Figure 2). CZ.ID (run in November of 2023) began with input validation and host depletion using STAR [65], followed by quality control steps such as adapter trimming with Trimmomatic [66], and low-quality read filtering using PRICE [67] and CD-HIT-DUP tool v4.6.8 [68]. The remaining reads were assembled into contigs with SPAdes [47] and then aligned to the NCBI nucleotide and protein databases through GSNAPL [69] and RAPsearch2 [70]. GD (run in December 2023) performed quality control using Trimmomatic and FastQC [71], followed by DIAMOND for protein-based alignment against viral databases. Groups of candidate viral reads were then assembled with metaSPAdes and the resultant contigs were then submitted to search against the references present in NCBI RefSeq using Blastn [72] and Blastx [55]. INSaFLU-TELEVIR’s workflow (run in December of 2024) also incorporated quality control with Trimmomatic and Prinseq++ [73] for read filtering. The reads were then enriched against two viral databases, Virosaurus90v 2020_4.2 (https://viralzone.expasy.org/8676, accessed on accessed on 6 December 2024) and NCBI refseq viral genomes release 4 (https://ftp.ncbi.nlm.nih.gov/genomes/refseq/, accessed on 6 December 2024) using Centrifuge, assembled with SPAdes and classified with the K-mer based tools Krakenuniq [74], Kraken2, Centrifuge and DIAMOND. Finally, confirmatory mapping was performed with Snippy, aligning the sequences against NCBI RefSeq Viral Genomes. The INSaFLU-TELEVIR reports are generated per workflow, meaning that, for each combination of software tools used, it can create a report. In this study, Krakenuniq, Kraken2, Centrifuge and DIAMOND were used, meaning the report obtained combined non-redundant hits detected across the four identification software tools.

We also used another simpler pipeline, without assembly, that consisted in processing the reads with Trimmomatic and Kraken 2 for its rapid taxonomic classification of raw reads without the need for assembly, allowing a complementary analysis alongside genome reconstruction approaches. This method is particularly useful for identifying a broad range of taxa while minimizing potential biases from assembly errors. Kraken2 used the database Prebuilt Refseq indexes: Viral (Version 7 June 2022—Downloaded: 4 August 2022. T105935Z, ran on the web-platform Galaxy Europe—https://usegalaxy.eu/—on 14 February 2024), a widely used tool, optimized for computational speed and efficiency. The results obtained were compared with those from CZ.ID, GD and INSaFLU-TELEVIR to validate consistency.

### 2.4. Automating the Analysis of Viral Identification

To streamline the analysis of viral identification reports from the four bioinformatic tools, a custom pipeline in Python was developed (Figure 3).

This script starts by reading each sample’s fastq.gz file to count reads and then consolidates information from all pipeline reports, storing read counts, organism names and taxonomic IDs in a data frame (df_taxid). The script filters out zero-read identifications (found in CZ.ID) and any identification flagged as being “low confidence” (found in INSaFLU—TELEVIR), and retrieves detailed taxonomic information for each viral identification using the latest ICTV taxonomy, automatically updated via Virus Metadata Resource (VMR). This taxonomy is organized in a directed graph format, allowing flexible identification even if taxonomic levels are missing. Each viral identification is tagged with a “full”, “partial” or “undefined” classification status, based on completeness. For example, viruses with classification up to the genus level but lacking family or order assignments are tagged as “partial”. The ETE toolkit (NCBITaxa class) is used for comprehensive taxonomic assignments, filling in lineage information from the kingdom to the species level. The assignment stops at the most specific available taxonomic level, as certain database entries lack full classification. Additionally, the pathogenic potential of each virus is flagged using a list derived from CZ.ID’s non-parsable pathogen list. Any taxa identified as belonging to a viral genus with potential human pathogens are marked accordingly, creating a refined pathogen identification dataset. A secondary data frame (table_samples) is created to summarize read counts across samples, including metadata such as the treatment plant number and collection details. This table enables downstream analysis by organizing data for each software pipeline and sample. The script then categorizes viral data into separate data frames: taxid_df_viruses for viruses, taxid_df_non_viruses for non-viral identifications, and further splits for phages and non-phages based on host-source filtering. This approach supports analysis specific to phages (e.g., those infecting bacteria or archaea) or broader viral classifications, following the ICTV host-source metadata. Taxa tables are generated for each viral category (e.g., non-viruses, viruses, non-phages, phages), organized for statistical analysis and visualization in R. Final outputs are saved in Excel files, including taxid_df.xlsx (unfiltered identifications) and Taxa tables for non-viruses, viruses, non-phages and phages. Additionally, a separate file stores entries without taxonomic information.

We used the script to analyze a total of 128 reports (8 sampling points, each with 3 replicates and 1 pooled sample across 4 tools), naming each report according to its sample and tool (e.g., A1_CZID). Options in the script allow users to update the NCBI taxonomy, adjust classification completeness parameters and define phage host categories. CrAssphage, a marker for human fecal contamination, is specifically tracked by filtering for the Crassvirales order. The full python script is available at (https://github.com/xiaodre21/comparison_of_viral_detection_methods, accessed on 21 December 2024).

### 2.5. Statistical Analysis

Descriptive statistic analysis was performed for the number of reads (raw, viral, phage and non-phage) and proportions were also computed.

To assess data distribution normality, Shapiro–Wilk tests were conducted. Due to deviations from normality, subsequent analyses employed non-parametric methods. Spearman’s correlation was used to examine relationships among quantitative variables. The Mann–Whitney test compared read counts between replicate and pooled samples, while the Kruskal–Wallis test assessed differences between software tools, followed by Dunn’s post hoc tests with Bonferroni correction for pairwise comparisons when Kruskal–Wallis results were significant.

For analyzing non-phage viral abundance, richness and diversity (alpha diversity) within samples, the R package Phyloseq was employed, utilizing number of observed unique taxa at species level (richness), Shannon and Simpson indices. Rarefaction curves were generated with the R package vegan to analyze the potential correlation between species richness and sampling effort across different sequencing tools. Beta diversity was calculated to evaluate between-sample diversity, visualized via Principal Coordinate Analysis (PCoA) based on Bray–Curtis dissimilarity to compare viral community compositions identified by each tool.

The effects of software pipeline, type of sample (replicate/pool), wastewater treatment plant and site (treatment step) on microbial community composition were analyzed using PERMANOVA (adonis2 package in R), a non-parametric multivariate analysis. All statistical analyses were conducted in R (version 4.3.0) using RStudio, with a significance level set at 0.05. The full R script is available online (https://github.com/xiaodre21/comparison_of_viral_detection_methods, accessed on 21 December 2024).

## 3. Results

### 3.1. Sequencing Performance and Viral Identification

A total of 32 samples (2 WWTPs, each with 4 sites, each with 3 replicates and 1 pool) were sequenced, and the number of raw reads ranged from 132,416 to 6,827,732, with a median of 906,068 and interquartile range (IQR) of 1,107,881. The RNA extraction concentrations of the replicate samples ranged from 61.14 to 114.03 ng/µL (Appendix A), with a median of 82.16 ng/µL and an IQR of 28.85 ng/µL. The pools of replicates were standardized to have an RNA concentration of 50 ng/µL. The Spearman correlation analysis revealed a positive and significant correlation between the number of raw reads and the concentration of nucleic acids (rho = 0.34, df = 89, *p*-value = 0.052). Considering the two groups, one with replicates and one with the pools, no statistically significant difference was found between them in the distributions of the numbers of raw reads (median of 906,068 and IQR of 978,000 in the replicates, and median of 881,004 and IQR of 1,129,542 in the pooled samples) (Mann–Whitney test, U= 76, *p*-value = 0.40).

From the total of 128 sample–software reports (full table in Appendix A), only 121 were retained. The numerical discrepancy arises because some samples either lacked any identification (for example, F3 from CZ.ID) or all identifications for a sample were flagged as having low confidence. This is why certain INSaFLU sample–software reports (B5, C5, D2, E5, F1, H1) are missing. The number of viral reads ranged from 3 to 288,464, with a median of 287 and an IQR of 1437. A positive correlation was also found between the number of raw reads and the number of viral reads identified with the four software tools, with a significant correlation for CZ.ID, GD and Trimmomatic + Kraken2 (Spearman correlation rho = 0.529, *p* = 0.002; rho = 0.572, *p* < 0.001; rho = 0.422, *p* = 0.016, respectively) and not for INSaFLU—TELEVIR (Spearman correlation, rho = 0.332, *p* = 0.097).

Viral read proportions relative to total raw reads ranged from 0.0003% (sample D3 on INSaFLU-TELEVIR) to 31.19% (D1 (WWTP1—Step 4) on GD). There was a significant difference in assigned viral read numbers between the four software tools (Kruskal–Wallis test, chi squared = 38.54, df = 3, *p* < 0.0001) with pairwise comparisons revealing significant differences between Trimmomatic + Kraken2 (median = 77, IQR = 186) and the other three software tools (CZ.ID—median = 1705, IQR = 2138; GD—median = 761, IQR = 1140; INSaFLU-TELEVIR—median = 56, IQR = 144; respectively), as well as between GD and INSaFLU–TELEVIR (Appendix A).

### 3.2. Phage and Non-Phage Viral Reads

The viral identifications were categorized as phage and non-phage, and again there was a wide range of values for the proportions of these categories among samples (Figure 4), from 100% phage reads (samples C5 and C1 (WWTP1—Step 3) analyzed with GD and INSaFLU-TELEVIR, respectively) to 100% non-phage reads (samples A5 (WWTP1—Step 1), B1 and B5 (both from WWTP1—Step 2)) analyzed with INSaFLU-TELEVIR. Although some samples had similar phage percentages across replicates and pools (namely, sample A (WWTP1—Step 1) and B (WWTP1—Step 2) in CZ.ID), the majority of the samples did not show this congruence (Figure 4). GD identified the largest percentage of phages (median = 57%), followed by Trimmomatic + Kraken2 and CZ.ID with a median percentage of 53% and 49%, respectively. For INSaFLU-TELEVIR this percentage was less than 1%, since the pipeline excludes phage identifications from the output reports. INSaFLU-TELEVIR identified the highest proportion of fully identified non-phage viruses, while CZ.ID identified most phage and non-phage viruses with only partial taxonomical resolution (Figure 4).

### 3.3. Viral Abundance and Diversity

Considering only the non-phages, the rarefaction curves (Figure 5A) for almost all samples showed an exponential increase in the number of taxa with increasing sample size (number of reads) but almost never reaching a plateau, meaning the sequence effort was likely not extensive enough, and that more taxa would probably be identified if more sequencing reads had been produced. The number of identified viral reads ranged from 10 to 4955. There was also a large variation in the non-phage viral total abundance of reads across software tools, but also across replicate and pool samples Figure 5B) at each treatment step of each WWTP.

The alpha diversity metrics (namely species richness), as well as Shannon and Simpson indices, showed a large variation between software tools (Figure 6). There were significant differences in these indices between the software groups (Kruskal–Wallis tests, observed: *p*-value = $1.435\times 10^{-10}$ Shannon: *p*-value = $1.391\times 10^{-9}$ and Simpson: *p*-value = $2.38\times 10^{-7}$), with INSaFLU-TELEVIR showing generally the lowest diversity values (Figure 6; post hoc tests in Appendix A).

The beta diversity analysis with PCoA revealed no clear distinction in viral community composition between software tools, WWTP, treatment step or type of sample (replicate/pools) (Figure 7A,B).

However, PERMANOVA tests using these factors revealed that the “software pipeline” factor has a significant effect on the overall viral composition (14.25% of the variation; F = 6.45, df = 3, *p* < 0.001). In contrast, “treatment plant” contributes only 1% to the variation and is not statistically significant (F = 1.50, df = 1, *p* = 0.06). The “treatment step” explains about 3.1% of the variation, with statistical significance (F = 1.41, df = 3, *p* = 0.017). The factor “Pool_Replicate” explains 2.0% of the variation, with statistical significance (F = 2.77, df = 1, *p* < 0.001). Most of the variation (79.47%) remains unexplained by these factors (Appendix A).

### 3.4. Pathogenic Viruses

Analyzing the reads from viruses which are potentially pathogenic to humans, 15 genera were identified, from eight different families, namely, Adenoviridae, Caliciviridae, Orthoherpesviridae, Papillomaviridae, Parvoviridae, Picornaviridae, Polyomaviridae and Sedoreoviridae (Appendix A). The number of reads from potentially pathogenic viruses in each sample ranged from 10 to 208 with a median value of 20. Again, there was a general lack of consistency between software tools, and also between replicates and pools (Figure 8). Some of the families were detected in all samples, at least by one of the software tools (e.g., *Caliciviridae* and Sedoreoviridae). Sample–software pairs with higher numbers of viral families with pathogenic members (e.g., replicate 1 of WWTP1—Step 3; pool of WWTP 2—Step 2; pool of WWTP2 Step 4, all with CZ.ID) were also the ones with higher viral read abundance (Figure 5).

CrAssphage, a widely used fecal contamination marker, occurred in 66 samples, including some of the later steps of each WWTP, with a number of reads ranging from 1 to 428 and with a median of 5 (Figure 8). The correlation between the number of crAssphage reads and the number of reads assigned to genera with potentially (human) pathogens revealed a positive and statistically significant correlation (rho = 0.71, *p* < 0.001). The correlations between crAssphage and either polyomaviruses, adenoviruses or bocaviruses was tested and the only statistically significant one was the first one (rho = 0.43, *p* < 0.0001 for polyomavirus, rho = 0.29, *p* = 0.001 for adenoviruses and rho = 0.47, *p* < 0.0001 for bocaviruses).

## 4. Discussion

This study compared viral sequences from triplicate samples and their pools from two wastewater treatment plants at four distinct treatment steps, analyzed with four bioinformatic software tools, aiming to analyze the consistency in viral detection and classification. Across the 128 sample–software reports, there was a lack of consistency of reported viral abundance and diversity between software tools and between replicates and their pools. Several factors can contribute to explain these findings, related to wastewater sample processing, laboratory procedures and bioinformatic analyses.

Wastewater samples are very complex matrices harboring many potential degradation and inhibition factors that may interfere in the performance/quality of the nucleic acid extraction, amplification and library preparation procedures. The large difference that we found in sequencing output (number of raw reads and number of viral reads) between the four samples (three biological replicates and the pool) in most sites may be due to those factors. Also, the heterogeneity of these complex matrices likely contributes to the large variation between biological replicates, casting doubt on the representativeness of sampling. Several studies on wastewater use composite samples, taken during a period of some hours [43], which can be considered as a pool of sequential samples during that time. Other studies analyze individually collected samples (grab samples), sometimes replicated. In our case, we collected replicates of grab samples, taken minutes apart, but, in some studies, the samples are collected a few days apart [75]. Within-site reproducibility is rarely taken into account, and there are few studies reporting viral sequencing results for each replicate. Similarly to our study, Schaeffer et al. [76], found variability in the number of raw reads and viral reads between four replicates taken from each wastewater sample collected in six WWTPs in northwestern France. Hendriksen et al. [77] collected replicates samples two days apart to analyze the reproducibility of the resistome based on metagenomic analysis of sewage. They reported higher similarity within than between sites. The choice of the volume of wastewater to process is a critical factor, since a large volume is needed to increase viral representativeness, but the concentration methods lead to increased inhibitor concentration. Here, we used 1 L of influent and 10 L wastewater at the three subsequent processing steps, to compensate for the expected higher concentration of viruses and also inhibition factors in the influent step in the former. However, we also need to consider that the treatment steps may introduce other inhibitory components (e.g., organic compounds, detergents and a plethora of other chemicals). Additionally, it is known that there are biases which can be introduced during the PCR amplification step, meaning the most abundant genomes are amplified while less abundant genomes can be under-represented or not even sequenced [78]. Wastewater sample processing can be performed with different methods that might influence the proportion of viral particles or genetic material in the sample. Zhang et al. [79] obtained a much larger proportion of metagenomic reads aligning with viral contigs (22.1% to 97.4%) when using the viral-like particle-concentrated method (VPC), with a size fractionation which recovers mostly free viral-like particles, than when using a non-concentration method (NC) that recovers the cellular fraction (0.9% to 17.4% of viral reads). A higher proportion of eukaryotic viruses in relation to phages was also detected in VPC metagenomes (19.4%) compared to NC metagenomes (4.3%) [62].

The quantity and quality of nucleic acid extractions also determines the quality of sequencing. In this study, there was a positive but non-significant correlation between the number of raw reads and the nucleic acid extraction concentration. It should be noted that we used a viral RNA extraction kit, but it also extracts DNA, allowing us to analyze both RNA and DNA viruses. We also have to consider that the quality ratios for the RNA extractions were generally low. For a 260/280 ratio, the values should ideally sit between 1.8 and 2.0 while the extracts prepared in the course of this work had 260/280 ratios between 2.28 and 3.09. For a 260/230 ratio, the values should be above 2.0 and they were between 0.18 and 0.66. However, this last ratio is known to be reduced most probably due to contamination with guanidine thiocyanate present in the lysis buffer of the extraction kit, which does not usually affect downstream procedures [80]. While the SISPA protocol and the VirCapSeq-VERT capture panel that were applied subsequently to the obtained nucleic acid extracts successfully allowed the enrichment and amplification of viral sequences, differences in the quality of the initial nucleic acid extraction might still have impacted the obtained results. Furthermore, the randomness of the SISPA amplification is also another factor that can influence the sequencing output. The lower specificity and sensitivity of this untargeted approach compared to targeted approaches, competition among abundant sequences and amplification biases all contribute to the challenges associated with untargeted amplification in viral metagenomics [25].

Pooled wastewater surveillance is a cost-efficient approach and has been implemented to monitor SARS-CoV-2, providing an efficient method for early virus detection [81]. Ray et al. [82] and Ko et al. [83] have shown that pooling samples before sequencing produces reliable estimates of diversity and effectively allows the identification of key biological signals comparable to individual sample sequencing. In our study, there was no clear pattern in the comparison of individual samples (replicates) with their pools, performed after the extraction of nucleic acids from the former with an equivalent concentration of each replicate. Overall, no significant differences in raw read counts were observed between pooled samples (median: 906,068 reads) and replicates (median: 881,004 reads). Nonetheless, for some samples, the number of raw reads and the viral abundance and diversity were higher in the pool, while for others it was the opposite, in some cases with large differences. These findings suggest that technical factors may influence sequencing outputs more than biological differences between pooled and individual samples. An approach to try to dilute the effect of technical randomness would be to sequence several replicates separately and merge the resulting raw reads for analysis [76,84].

We explored the impact of sequencing effort on viral abundance and found a moderate positive correlation between the absolute number of raw reads and viral reads, suggesting that increased sequencing depth enhanced viral detection, as expected. Rarefaction curve analysis further indicated that the sequencing depth was insufficient to capture the full viral diversity, as most curves did not reach a plateau. These findings imply that we have an underestimation of viral diversity and that deeper sequencing could reveal additional viral species, including low-abundance pathogens. Rarefaction curves are rarely presented in viral metagenomic studies, and they are crucial to understand the validity of comparative analysis. For example, when a large variability in the absolute number of reads is observed between samples, it may be misleading to compare the viral diversity (e.g., number of families or species and their relative proportions), because this diversity will be affected by the total number of reads. Future studies should consider real-time viral detection during sequencing to optimize sequencing depth and run times for comprehensive pathogen surveillance.

The percentage of viral reads (in relation to the raw reads) in our study ranged from 0.0002% to 31.19%. Other metavirome studies conducted on wastewaters reported a wide range of values. Gulino et al. [29] found that 4.1% of the reads were viral but, from these, only 10% could be taxonomically assigned beyond “Virus”. These included mainly phages and a few eukaryotic viruses (0.006%). In Tamaki et al. [33], 14.8% to 21% of the sequences were taxonomically classified and, from these, 13.7% to 18.5% were assigned to viruses. Zhang et al. [79] found 0.9% to 97.4% of metagenomic reads aligning with viral contigs, depending on the sample processing methods. Rothman et al. [31] found that 9.1% to 43.3% were taxonomically classified, of which 0.65% to 13.4% were viral.

Different software tools yielded contrasting detections of viral read counts, with CZ.ID identifying the highest median viral reads and Trimmomatic + Kraken2 the lowest. The choice of software thus significantly affects outcomes, as each tool has specific algorithms and databases tailored to different detection objectives (e.g., broad metagenomic detection vs. viral sequence specificity). For example, phage and non-phage virus proportions varied significantly between software tools. INSaFLU-TELEVIR, designed to detect eukaryotic viruses, discards phage identifications, although a few are still detected. The completeness of taxonomic information also differed between software tools. CZ.ID displayed a higher proportion of partially classified reads for both phage and non-phage sequences, likely due to the breadth of its reference databases and alignment methods, which may struggle with divergent or under-represented viruses. GD effectively classified non-phage reads but showed more partially identified phage sequences, reflecting the genetic complexity and under-representation of phages in databases. INSaFLU-TELEVIR achieved the highest proportion of fully classified non-phage sequences, likely due to the use of a more curated database, which may have also lowered the alpha diversity metrics for this software tool. Trimmomatic + Kraken2, leveraging a k-mer-based approach, showed a moderate balance between partially and fully classified non-phage viruses. While efficient, its reliance on k-mer matching limits the performance for sequences with high genetic variability or poor database representation. While alignment-based methods are more resource-intensive, they provide robust solutions for complex analyses.

The four bioinformatic approaches used—Genome Detective, CZ.ID, INSaFLU-TELEVIR and Trimmomatic + Kraken2—each have distinct strengths and limitations. Genome Detective provides rapid and comprehensive viral identification and assembly, leveraging curated databases to reliably detect known viruses, but its use is subscription-based, potentially limiting accessibility. CZ.ID is freely available, user-friendly, and optimized for broad pathogen discovery and surveillance, yet its cloud-based nature may raise data security and privacy concerns. INSaFLU-TELEVIR specializes in surveillance and detailed genomic analyses of known viruses relevant to public health, but it inherently excludes bacteriophages and may lack sensitivity towards novel or divergent viruses in complex environmental samples. Trimmomatic + Kraken2 offers fast, k-mer-based taxonomic assignment suitable for high-throughput datasets without extensive computational resources; however, its reliance on database completeness can limit sensitivity and specificity, particularly in detecting novel or highly divergent viruses. Therefore, the choice of analytical method should be guided by study objectives, resource availability, desired specificity versus breadth and data privacy considerations. The lack of consensus between software tools underlines the potential need for the use of various tools, as the complementarity between them could provide a more comprehensive overview of the results. We provide a pipeline for summarizing and analyzing the reports from these several tools, extendable to other tools.

Due to the observed constraint of insufficient sequencing effort in our samples, the comparison between WWTPs and treatment steps should be taken with caution. However, considering the available data, we were able to report 35 viral families in wastewater samples, showing persistence in viral abundance and diversity along the wastewater treatment steps. Inefficiency of WWTPs in removing viruses was also found in other studies, e.g., in domestic municipal WWTPs in Singapore [33] and in municipal, swine and duckery WWTPs in China [79] and others [11,12,29,85]. In our study, WWTP1 was experiencing a malfunction of the hypochlorite step, which may have compromised the microbial reduction even further.

Replicates of the same sample appeared scattered rather than grouped in the PCoA analysis, reflecting the lack of consistency in viral composition between replicates, as already suggested by the differences seen in the analyses of the sequencing output and alpha diversity. This inconsistency between replicates is probably due, as discussed already, to the inherent heterogeneity of the wastewater matrices. PERMANOVA analysis indicated that the choice of software significantly influenced viral composition patterns, whereas factors like WWTP and collection steps explained only a small proportion of the variation.

Eight families of potential human viral pathogens were identified, including those causing respiratory infections, gastroenteritis, hepatitis and even virus-inducing cancers (e.g., alphapapillomaviruses). These findings highlight the value of wastewater monitoring for public health, as it can provide insights into pathogen transmission within communities. Consistent with previous studies, families such as *Adenoviridae*, *Orthoherpesviridae*, *Papillomaviridae* and *Parvoviridae* were detected, aligning with the findings of wastewater and animal farm samples [36,86] and studies on sewage-detected viruses [1,87]. Key genera of public health importance included *Mastadenovirus* (some serotypes causing respiratory infections and gastroenteritis), Norovirus and Sapovirus (leading causes of viral gastroenteritis), *Enterovirus* and *Hepatovirus* (associated with polioviruses and hepatitis A), and *Rotavirus* (a major cause of severe diarrhea in children) [88]. The primary transmission route for these viruses is fecal–oral, emphasizing the importance of wastewater surveillance, particularly in areas with poor sanitation [89]. This diversity of pathogens reinforces the need for comprehensive monitoring to address multiple health risks. The study demonstrated the utility of metagenomics in viral surveillance, identifying clinically significant viruses, and suggesting wastewater-based epidemiology as a valuable tool to track infectious diseases. Early detection of viruses like norovirus or enterovirus through wastewater can serve as a warning system for outbreaks. Surveillance of other water sources (e.g., drinking or seawater) could complement these efforts, although current EU and Portuguese regulations focus solely on fecal coliform bacteria. The recent EU directive 2024/3019 already establishes the monitoring of viruses to be included in national systems for urban wastewater surveillance (https://eur-lex.europa.eu/eli/dir/2024/3019/oj, accessed on 5 April 2025).

CrAssphage is used as a marker of human fecal pollution and a proxy for pathogenic viruses [90,91,92] making it a useful indicator of viral load and a potential measure of treatment efficacy in wastewater plants [93,94]. In this study, correlations between crAssphage and human pathogenic viruses were moderate and significant, particularly for polyomaviruses and adenoviruses, further emphasizing the potential of crAssphage as a fecal contamination marker and as a tool in public health surveillance. Again, its detection at all WWTP treatment steps reveals the inefficiency of viral elimination in WWTPs.

Based on our results, we may suggest some approaches to improve the resilience and robustness of wastewater virome analyses. Composite or increased replicate sampling can enhance representativeness, mitigating the inherent heterogeneity of wastewater samples. Optimizing concentration methods to reliably enrich viral particles and reduce inhibitory compounds, alongside standardized nucleic acid extraction protocols specifically designed for environmental matrices, is critical. Moreover, sequencing depth should be sufficient to capture viral diversity, including low-abundance viruses. The concurrent application of multiple bioinformatic tools could further strengthen viral detection and classification. Finally, the development of standardized guidelines and inter-laboratory validation studies will help ensure consistency and comparability across wastewater surveillance efforts.

In this study, we focused on the identification of known viruses. However, adopting a metagenomics approach for studying viruses in wastewater, combined with advanced analysis tools for viral discovery [95] and pathogen potential prediction [96], would significantly enhance surveillance programs, contributing to improved pandemic preparedness.

Future work could focus on developing and validating standardized protocols for wastewater sampling, concentration, nucleic acid extraction and amplification. This would help mitigate the variability observed between replicates and reduce biases introduced by different processing methods. For example, comparing composite sampling versus grab sampling and refining inhibitor removal techniques could enhance consistency. At the same time, we believe that incorporating machine learning algorithms might also improve the sensitivity and specificity of viral detection and taxonomic classification, thereby enhancing reproducibility across different studies. Further research could explore the integration of virome data with epidemiological and environmental information. Investigating correlations between viral markers (such as crAssphage) and the prevalence of human pathogens could provide valuable insights for public health risk assessments and guide the development of more effective intervention strategies.

## 5. Conclusions

The wide variability observed across individual replicates and pooled samples highlights the intrinsic heterogeneity of wastewater matrices and the importance of technical consistency in sample processing, extraction, PCR amplification and sequencing. We also identified strengths and limitations of bioinformatic software tools for characterizing the viral taxonomic profiles and provided a Python script to automate the analysis of output reports from each tool, allowing the integration of results that can be relevant for wastewater surveillance programs.

## Figures and Tables

**Figure 1 ijerph-22-00707-f001:**
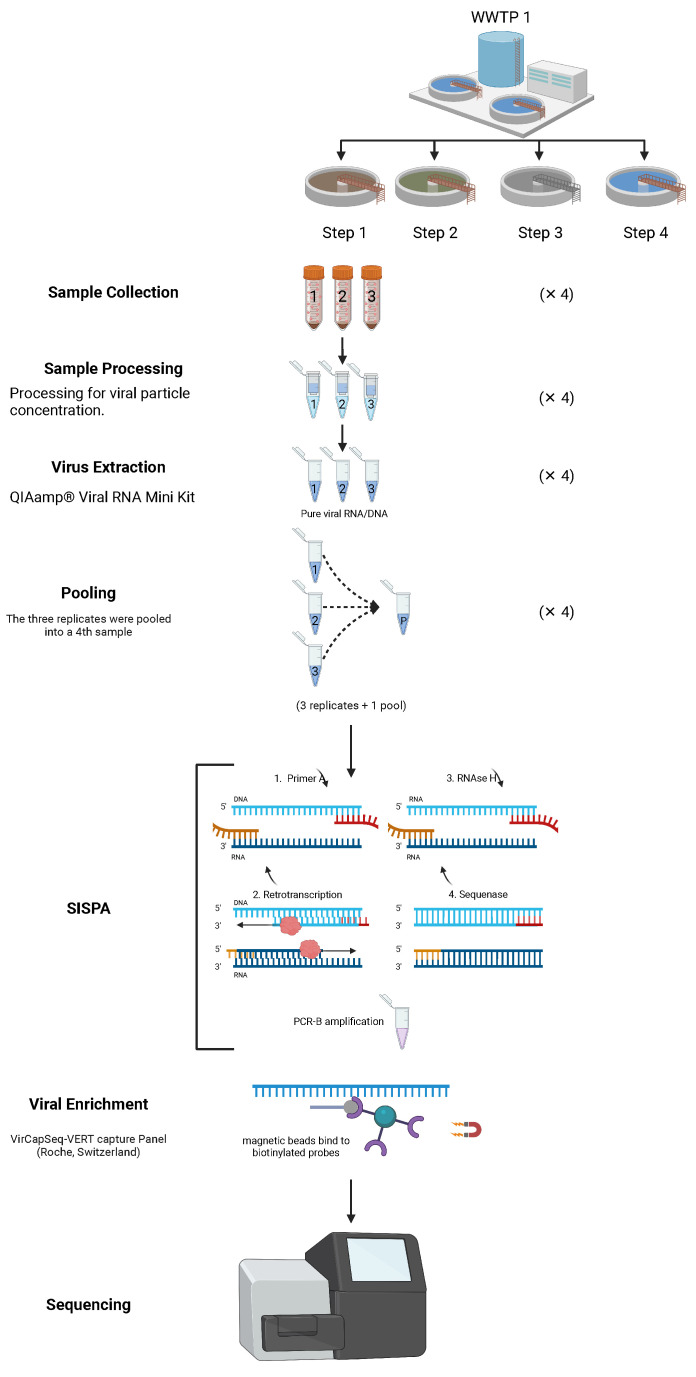
Schematic representation of the sampling and laboratory procedures used in the study of wastewater samples. SISPA and viral enrichment methods adapted from [60] created in Biorender by Santos, A in 2025 https://BioRender.com/i02a476, accessed on 4 March 2025.

**Figure 2 ijerph-22-00707-f002:**
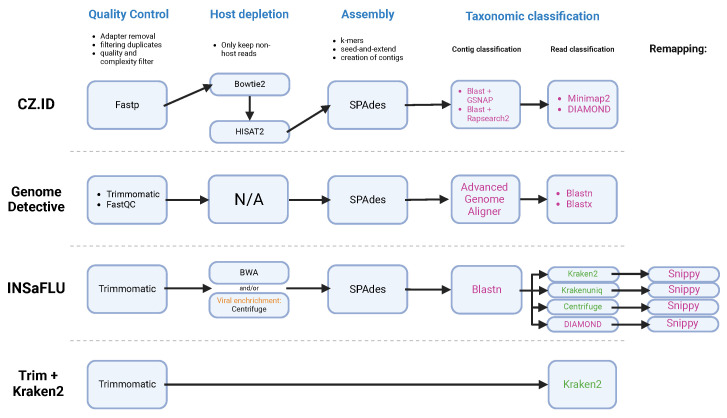
Diagram of the workflows of each bioinformatic pipeline, depicting the steps (in blue) of quality control, host depletion, assembly and taxonomic classification. CZ.ID, GD, TELEVIR module of INSaFLU and Trimmomatic + Kraken2 ran on Galaxy Community Hub [74]. The software tools used for taxonomic classification are colored according to the method used: (pink) alignment-based; (green) k-mer.

**Figure 3 ijerph-22-00707-f003:**
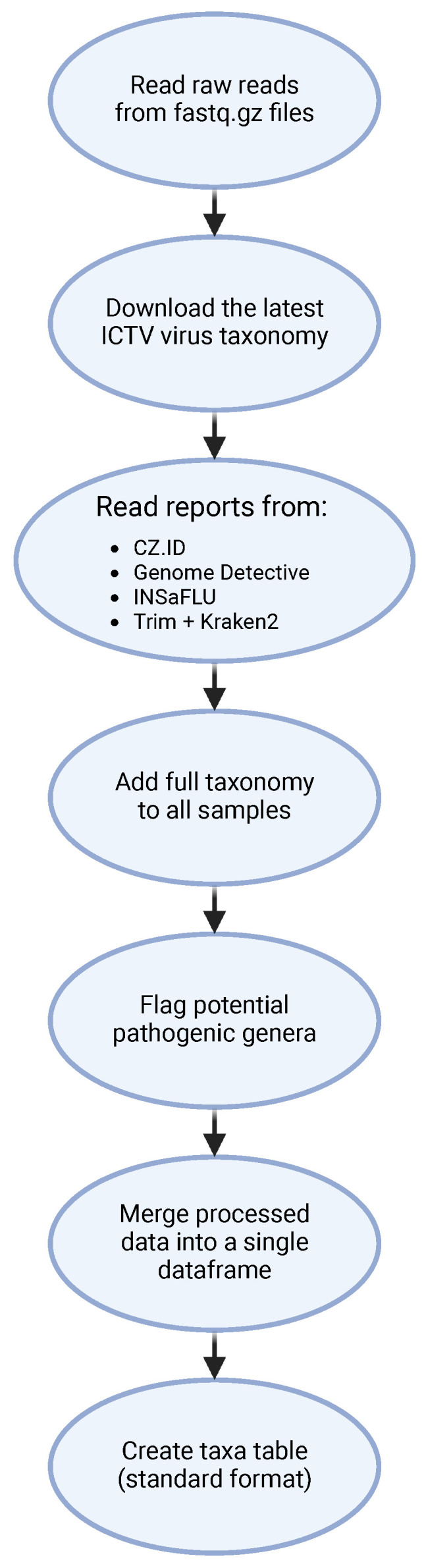
Fluxogram of the Python script for processing and analyses of the reports generated by the different software pipelines. Taxa—Operational Taxonomic Unit.

**Figure 4 ijerph-22-00707-f004:**
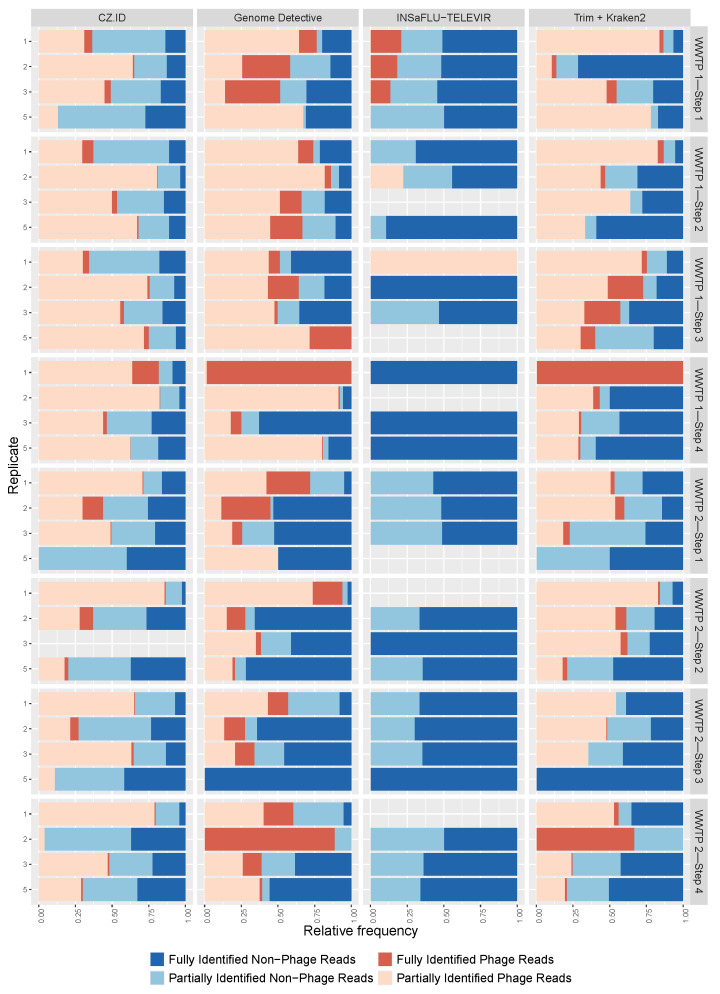
Proportion of phage (shades of red) and non-phage (shades of blue) for each sample type (replicate 1, 2, 3 and pools—5), by software (columns) and treatment step for each wastewater treatment plant (WWTP) (rows). Dark colors represent the full identifications (complete taxonomical resolution from Superkingdom to Species) and the light colors represent the partially identifications (at least one taxonomic rank missing from Superkingdom to Species). Absence of color indicates absence of viral reads.

**Figure 5 ijerph-22-00707-f005:**
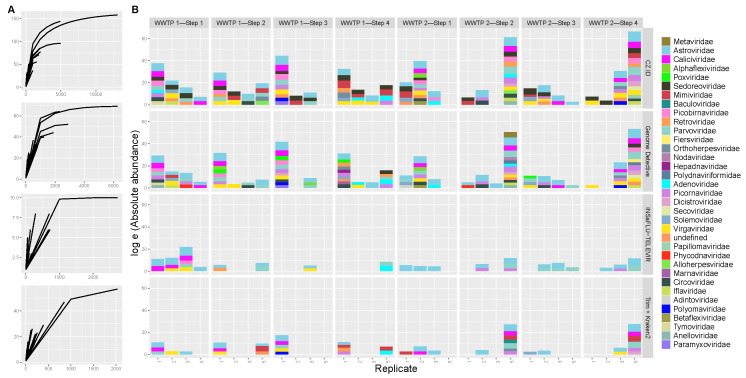
(**A**) Rarefaction curves showing the number of identified taxa (y-axis) increasing with higher number of reads (x-axis) for each wastewater sample (each curve), for each software tool (rows). The sample labels of the curves are shown in Appendix A. (**B**) Stacked barplot for log absolute frequency of reads for each sample type (replicate 1, 2, 3 and pools—5), by software (rows) and treatment step for each wastewater treatment plant (WWTP) (columns) by viral family (colors).

**Figure 6 ijerph-22-00707-f006:**
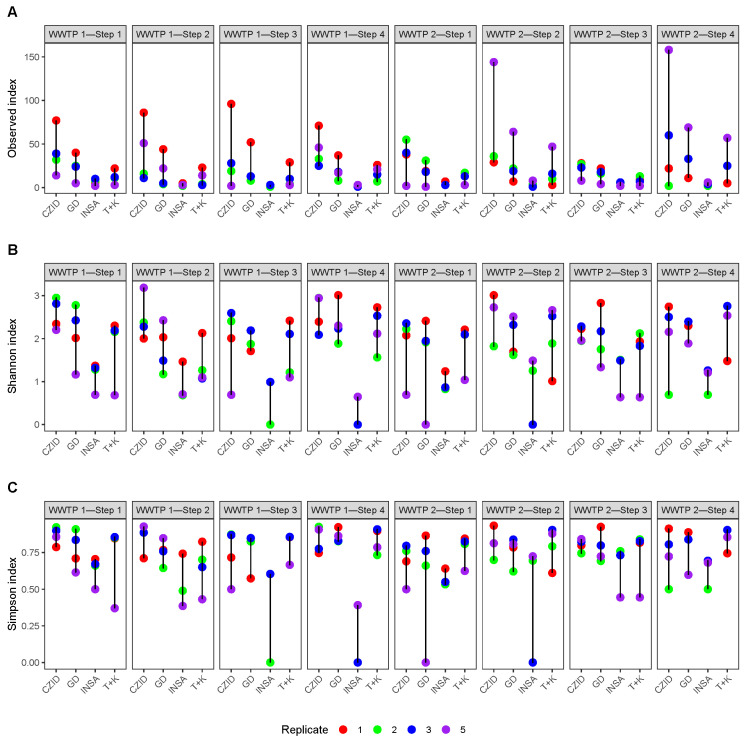
Comparison of the alpha diversity metrics across the different software tools (x-axis) and collection steps (top legends), by replicate (1, 2 and 3) and pool (5). (**A**) Observed number of taxa; (**B**) Shannon Index; (**C**) Simpson Index. Labels for the x-axis: CZID = CZ.ID; GD = Genome Detective; INSA = INSaFLU-TELEVIR; T + K = Trim + Kraken2.

**Figure 7 ijerph-22-00707-f007:**
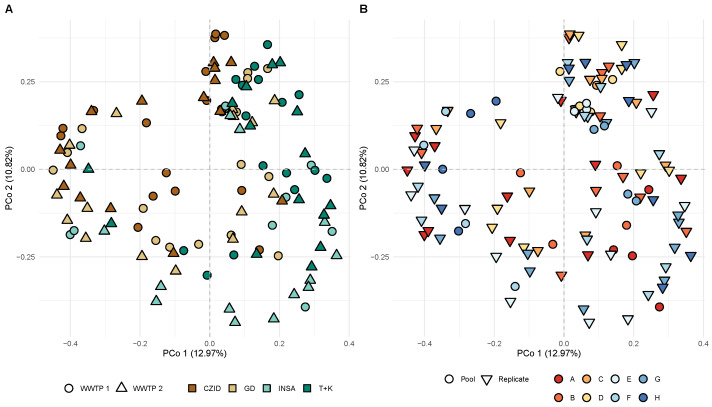
Principal Coordinate Analysis (PCoA) based on the Bray–Curtis dissimilarity matrix. The same scatter plot of PCo1 versus PCo2 is shown with colors and symbols discriminating by (**A**) WWTP (wastewater treatment plants 1 and 2) and software tools (CZ.ID, GD, INSaFLU-TELEVIR, Trimmomatic + Kraken2); (**B**) sample type (pool or replicate) and sample of each treatment step at each WWTP (A–H). Labels for the x-axis: CZID = CZ.ID; GD = Genome Detective; INSA = INSaFLU-TELEVIR; T + K = Trim + Kraken2. The dashed lines indicate the origin of the x- and y-axes.

**Figure 8 ijerph-22-00707-f008:**
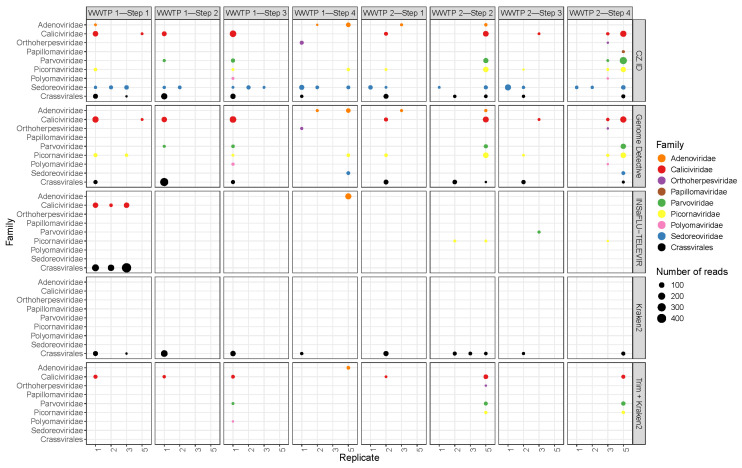
Bubble plot of abundance (number of reads) of potentially pathogenic viruses and of Crassvirales across all the samples (columns) and software tools (rows).

## Data Availability

Python and R scripts are available in https://github.com/xiaodre21/comparison_of_viral_detection_methods, accessed on 21 December 2024. Raw sequencing files can be found in NCBI’s BioProject: PRJNA1246641, https://www.ncbi.nlm.nih.gov/bioproject/PRJNA1246641, accessed on 4 April 2025.

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
