# Peer review of "Wastewater Metavirome Diversity: Exploring Replicate Inconsistencies and Bioinformatic Tool Disparities"

_ijerph, 2025, doi:10.3390/ijerph22050707_

Round 1

Reviewer 1 Report

Comments and Suggestions for Authors

This study is comprehensive and technically sound. I have queries as below:

What suggestions are there to increase the virome analyses' resilience in wastewater?

After assessing the effectiveness of four bioinformatics tools, which one does the author suggest other researchers use?

Author Response

Comments 1: What suggestions are there to increase the virome analyses' resilience in wastewater?

Response 1: We thank the reviewer for this insightful question. Enhancing the resilience of wastewater virome analyses is indeed challenging given the complexity and novelty of the field. We have added a new paragraph in the discussion summarizing the best practices for a resilient approach to wastewater virome analyses:

“Based on our results, we may suggest some approaches to improve the resilience and robustness of wastewater virome analyses.  Composite or increased replicate sampling can enhance representativeness, mitigating the inherent heterogeneity of wastewater samples. Optimizing concentration methods to reliably enrich viral particles and reduce inhibitory compounds, alongside standardized nucleic acid extraction protocols specifically designed for environmental matrices, is critical. Moreover, sequencing depth should be sufficient to capture viral diversity, including low-abundance viruses. The concurrent application of multiple bioinformatics tools could further strengthen viral detection and classification. Finally, the development of standardized guidelines and inter-laboratory validation studies will help ensure consistency and comparability across wastewater surveillance efforts.” (Lines 505 to 514 in the new version of the manuscript.)

Comments 2: After assessing the effectiveness of four bioinformatics tools, which one does the author suggest other researchers use?

Response 2: We appreciate the reviewer’s comment.  Given that each tool serves distinct purposes and involves practical considerations, we refrain from recommending one universally superior option. Instead, we have now included a sum-up of strengths and limitations of each tool and guidelines to select bioinformatics tools based on their specific research objectives, data type, and practical constraints:

“The four bioinformatic approaches used—Genome Detective, CZ.ID, INSaFLU-TELEVIR, and Trimmomatic+Kraken2—each have distinct strengths and limitations. Genome Detective provides rapid and comprehensive viral identification and assembly, leveraging curated databases to reliably detect known viruses, but its use is subscription-based, potentially limiting accessibility. CZ.ID is freely available, user-friendly, and optimized for broad pathogen discovery and surveillance, yet its cloud-based nature may raise data security and privacy concerns. INSaFLU-TELEVIR specializes in surveillance and detailed genomic analyses of known viruses relevant to public health, but it inherently excludes bacteriophages and may lack sensitivity towards novel or divergent viruses in complex environmental samples. Trimmomatic+Kraken2 offers fast, k-mer-based taxonomic assignment suitable for high-throughput datasets without extensive computational resources; however, its reliance on database completeness can limit sensitivity and specificity, particularly in detecting novel or highly divergent viruses. Therefore, the choice of analytical method should be guided by study objectives, resource availability, desired specificity versus breadth, and data privacy considerations.”(Lines 441 to 455 in the new version of the manuscript).

Reviewer 2 Report

Comments and Suggestions for Authors

This manuscript explores viral diversity in wastewater using metagenomics, with a focus on inconsistencies between replicates and the influence of different bioinformatic tools on viral detection and classification. The introduction provides a clear rationale for the study, situating it within the broader context of wastewater surveillance for public health.

The methodology is well described, including sample collection from multiple wastewater treatment plant sites, processing of replicates and pooled samples, and analysis using various bioinformatics pipelines. The use of sequence-independent single-primer amplification (SISPA) and viral enrichment methods strengthens the technical approach.

The results reveal notable variability in viral abundance and diversity across replicates and bioinformatic tools, highlighting the complexity of wastewater samples and the challenges in ensuring reproducibility in viral metagenomic studies. The findings suggest that sequencing depth and bioinformatics tool selection can significantly influence viral detection, emphasizing the importance of standardized protocols to improve data consistency.

Overall, this study offers valuable insights into the application of metagenomics in wastewater-based epidemiology. These findings contribute to ongoing efforts to enhance viral monitoring in wastewater and will be of interest to researchers in environmental microbiology, public health, and water treatment.

Specific comments

Line 105: What does the “treatment steps” specifically refer to?

Line 105: Please delete a period after “(Figure 1)”.

Line 278: There are a total of 128 sample software reports, but only 121 are displayed in the table.

Line 315-316: Please indicate A and B in Fig. 5.

Line 354: Could you explore the comparative strengths, limitations, and distinguishing characteristics of the four methodological approaches (i.e., Genome Detective, CZ.ID, INSaFLU-TELEVIR and Trimmomatic+Kraken2) employed in Discussion section?

Author Response

Comments 1: Line 105: What does the “treatment steps” specifically refer to?

Response 1: We thank the reviewer for highlighting this point. We have now added this explanation in the methods section:  “"Treatment steps" refers specifically to distinct stages within each wastewater treatment plant (WWTP), namely: 1) Influent (raw wastewater entry), 2) Post-primary treatment (solids and sediments removal), 3) Secondary treatment (biological treatment with microbial processes), and 4) Effluent (final treated wastewater released into the environment).” (lines 91 to 95 in the new version of the manuscript). .

Comments 2: Line 278: There are a total of 128 sample software reports, but only 121 are displayed in the table.

Response 2: We thank the reviewer for the comment. The difference in numbers is due to the fact that either the samples did not have any identification (F3 from CZ.ID) or due to all identifications, for a specific sample, being flagged for a low confidence level. This is the reason why some INSaFLU sample software reports are missing (samples B5, C5, D2, E5, F1, H1). We have included this explanation in the methods and in the results.

“This script starts by reading each sample’s fastq.gz file to count reads and then consolidates information from all pipeline reports, storing read counts, organism names, and taxonomic IDs in a data frame (df_taxid). The script filters out zero-read identifications (found in CZ.ID) and any identification flagged as being “low confidence” (found in INSaFLU - TELEVIR) and retrieves detailed taxonomic information for each viral identification using the latest ICTV taxonomy, automatically updated via Virus Metadata Resource (VMR).” (lines 175 and 176 of revised manuscript).

“(…) only 121 were retained. The numerical discrepancy arises because some samples either lacked any identification (for example, F3 from CZ.ID) or all identifications for a sample were flagged as having low confidence. This is why certain INSaFLU sample software reports (B5, C5, D2, E5, F1, H1) are missing.” (lines 247 to 250 in the revised manuscript).

Comments 3: Line 354: Could you explore the comparative strengths, limitations, and distinguishing characteristics of the four methodological approaches (i.e., Genome Detective, CZ.ID, INSaFLU-TELEVIR and Trimmomatic+Kraken2) employed in Discussion section?

Response 3: We appreciate the reviewer’s suggestion and, also in accordance with reviewer’s 1 suggestion, we have added a new paragraph in the discussion:

“The four bioinformatic approaches used—Genome Detective, CZ.ID, INSaFLU-TELEVIR, and Trimmomatic+Kraken2—each have distinct strengths and limitations. Genome Detective provides rapid and comprehensive viral identification and assembly, leveraging curated databases to reliably detect known viruses, but its use is subscription-based, potentially limiting accessibility. CZ.ID is freely available, user-friendly, and optimized for broad pathogen discovery and surveillance, yet its cloud-based nature may raise data security and privacy concerns. INSaFLU-TELEVIR specializes in surveillance and detailed genomic analyses of known viruses relevant to public health, but it inherently excludes bacteriophages and may lack sensitivity towards novel or divergent viruses in complex environmental samples. Trimmomatic+Kraken2 offers fast, k-mer-based taxonomic assignment suitable for high-throughput datasets without extensive computational resources; however, its reliance on database completeness can limit sensitivity and specificity, particularly in detecting novel or highly divergent viruses. Therefore, the choice of analytical method should be guided by study objectives, resource availability, desired specificity versus breadth, and data privacy considerations.” (Lines 441 to 455 in the new version of the manuscript).

Reviewer 3 Report

Comments and Suggestions for Authors

Comments to authors

I recommend reviewing the introduction, leaving only the necessary information, removing the unnecessary content, and relocating some of it to the discussion section.

The Materials and Methods section could be made more concise, and Figure 3 could be resized to fit the page. Also, the font size could be more suitable for reading.

I recommend revising the footnotes as follows: Schematic representation of the sampling and laboratory procedures used in the study of wastewater samples. SISPA and viral enrichment methods adapted from [60] Created in BioRender. Santos, A. (2025) instead of Schematic representation of the sampling scheme and of the laboratory procedures used for the study of wastewater samples from two Wastewater Treatment Plants (WWTP), in 4 Treatment Steps. At each step, three replicates (1, 2, and 3) were collected, processed, and sequenced individually, while a subsample of the nucleic acid extractions from these replicates was combined to create a pooled sample (P), which was also sequenced. SISPA and viral enrichment methods adapted from [60] Created in BioRender. Santos, A. (2025).

The conclusions should be revised to clearly and precisely address the paper's aims.

Ultimately, I believe that to add value to the results, the authors should provide recommendations for future work based on their findings.

Author Response

Comments 1: I recommend reviewing the introduction, leaving only the necessary information, removing the  

Response 1: We thank the reviewer for the suggestion We reviewed the introduction and removed or reduced some sections: we removed a section with details of percentage of reads assigned to viruses in other studies (lines 40 to 49 in the original version); we  removed the names of the softwares which are unnecessary in the introduction (lines 87 to 90 in the original version); and we reduced the paragraph on the methodological biases (lines 45 to 66)

"K-mer-based tools [50, 51] perform rapid matching of short sequence fragments to indexed reference databases, though they may lack sensitivity for novel or highly diverse sequences [53,54]. Alignment-based methods [55, 56, 57], are computationally demanding but offer greater precision by aligning sequences to known databases [58]. " (lines 74 to 80 in the revised manuscript)”

“For a more comprehensive analysis of the virome, an alternative is to use enrichment methods, for example with viral probes, and random primer amplification [24,36,4,37]. However, enrichment methods can distort viral abundance profiles [32,38–40], calling for standard protocols and benchmarking. Biases may also be introduced during sample collection and processing [38,41,42] . One primary bias arises from sampling methods, such as grab samples, which capture only a snapshot of the wastewater at a single time point and may not reflect variations in viral or microbial load throughout the day. Composite sampling over longer periods can mitigate this but introduces its own challenges, such as degradation of sensitive bio-molecules like RNA during collection or storage [43]. Another source of bias is heterogeneity in wastewater composition, influenced by population density, industrial discharge, and environmental factors like rainfall, which can dilute samples or introduce contaminants [44]. Sample handling and processing factors, for instance, inefficient concentration methods may preferentially recover some microorganisms over others [34]. Lastly, biases can stem from analytical techniques, such as  PCR amplification bias, or reagent contamination, which can introduce false  positives or obscure low-abundance taxa [45]. Mitigating these biases will imply careful sampling design, robust protocols, and appropriate controls. (lines 45 to 62 in the revised manuscript).”

Round 2

Reviewer 3 Report

Comments and Suggestions for Authors

Several changes are noted in the manuscript review; however, comments are not specifically addressed, making them difficult to identify. Therefore, I'm copying the comments again.

The Materials and Methods section could be made more concise, and Figure 3 could be resized to fit the page. Also, the font size could be more suitable for reading.

I recommend revising the footnotes as follows: Schematic representation of the sampling and laboratory procedures used in the study of wastewater samples. SISPA and viral enrichment methods adapted from [60] Created in BioRender. Santos, A. (2025) instead of Schematic representation of the sampling scheme and of the laboratory procedures used for the study of wastewater samples from two Wastewater Treatment Plants (WWTP), in 4 Treatment Steps. At each step, three replicates (1, 2, and 3) were collected, processed, and sequenced individually, while a subsample of the nucleic acid extractions from these replicates was combined to create a pooled sample (P), which was also sequenced. SISPA and viral enrichment methods adapted from [60] Created in BioRender. Santos, A. (2025).

The conclusions should be revised to clearly and precisely address the paper's aims.

Ultimately, I believe that to add value to the results, the authors should provide recommendations for future work based on their findings.

Author Response

We would like to express our appreciation towards the comments given and also apologize for the minor mistake as although the suggested changes were performed, the responses adressing the comments were not properly submitted. We hereby submit the answers to the comments left unadressed.

Comments 1: The Materials and Methods section could be made more concise, and Figure 3 could be resized to fit the page. Also, the font size could be more suitable for reading.

Response 1: Following this suggestion, we removed from the Materials and Methods the details about the viral particle concentration methods (lines 111 to 125 in the original version) and about the SISPA protocol (lines 143 to 156 in the original version) and we put them as Supplementary Material. The size of the figure and font size will be refit by production.

Comments 2: I recommend revising the footnotes as follows: Schematic representation of the sampling and laboratory procedures used in the study of wastewater samples. SISPA and viral enrichment methods adapted from [60] Created in BioRender. Santos, A. (2025) instead of Schematic representation of the sampling scheme and of the laboratory procedures used for the study of wastewater samples from two Wastewater Treatment Plants (WWTP), in 4 Treatment Steps. At each step, three replicates (1, 2, and 3) were collected, processed, and sequenced individually, while a subsample of the nucleic acid extractions from these replicates was combined to create a pooled sample (P), which was also sequenced. SISPA and viral enrichment methods adapted from [60] Created in BioRender. Santos, A. (2025).

Response 2: We have changed it as suggested.

Comments 3: The conclusions should be revised to clearly and precisely address the paper's aims.

Response 3: Reply: Thank you for the suggestion. We have now revised the conclusions accordingly:

“The wide variability observed across individual replicates and pooled samples, highlights the intrinsic heterogeneity of wastewater matrices and the importance of technical consistency in sample processing, extraction, PCR amplification and sequencing. We also identified strengths and limitations of bioinformatic software tool for characterizing the viral taxonomic profiles and provided a Python script to automate analysis of output reports from each tool, allowing the integration of results that can be relevant for wastewater surveillance programs." .”(Lines 519 to 525 in the new version of the manuscript).

Comments 4: Ultimately, I believe that to add value to the results, the authors should provide recommendations for future work based on their findings.

Response 4: We appreciate the reviewer’s suggestion and we have added a new paragraph in the discussion:

“Future work could focus on developing and validating standardized protocols for wastewater sampling, concentration, nucleic acid extraction, and amplification. This would help mitigate the variability observed between replicates and reduce biases introduced by different processing methods. For example, comparing composite sampling versus grab sampling and refining inhibitor removal techniques could enhance consistency. At the same time, we believe that incorporating machine learning algorithms might also improve the sensitivity and specificity of viral detection and taxonomic classification, thereby enhancing reproducibility across different studies.

Further research could explore the integration of virome data with epidemiological and environmental information. Investigating correlations between viral markers (such as crAssphage) and the prevalence of human pathogens could provide valuable insights for public health risk assessments and guide the development of more effective intervention strategies.” (Lines 519 to 530 in the new version of the manuscript).
